# Rheological and Morphological Properties of Oil Palm Fiber-Reinforced Thermoplastic Composites for Fused Deposition Modeling (FDM)

**DOI:** 10.3390/polym13213739

**Published:** 2021-10-29

**Authors:** Mohd Nazri Ahmad, Mohamad Ridzwan Ishak, Mastura Mohammad Taha, Faizal Mustapha, Zulkiflle Leman

**Affiliations:** 1Department of Aerospace Engineering, Faculty of Engineering, Universiti Putra Malaysia, Serdang 43400, Selangor, Malaysia; faizalms@upm.edu.my; 2Faculty of Mechanical and Manufacturing Engineering Technology, Universiti Teknikal Malaysia Melaka, Hang Tuah Jaya, Durian Tunggal 76100, Melaka, Malaysia; mastura.taha@utem.edu.my; 3Centre of Smart System and Innovative Design, Universiti Teknikal Malaysia Melaka, Hang Tuah Jaya, Durian Tunggal 76100, Melaka, Malaysia; 4Aerospace Malaysia Research Centre (AMRC), Universiti Putra Malaysia, Serdang 43400, Selangor, Malaysia; 5Laboratory of Biocomposite Technology, Institute of Tropical Forestry and Forest Products (INTROP), Universiti Putra Malaysia, Serdang 43400, Selangor, Malaysia; 6Department of Mechanical and Manufacturing Engineering, Faculty of Engineering, Universiti Putra Malaysia, Serdang 43400, Selangor, Malaysia; zleman@upm.edu.my; 7Advanced Engineering Materials and Composites Research Centre, Faculty of Engineering, Universiti Putra Malaysia, Serdang 43400, Selangor, Malaysia

**Keywords:** FDM, oil palm fiber, ABS, rheology

## Abstract

Fused deposition modelling (FDM) is a filament-based rapid prototyping technology that allows new composite materials to be introduced into the FDM process as long as they can be manufactured in feedstock filament form. The purpose of this research was to analyze the rheological behavior of oil palm fiber-reinforced acrylonitrile butadiene styrene (ABS) composites when used as a feedstock material, as well as to determine the best processing conditions for FDM. The composite’s shear thinning behavior was observed, and scanning electron microscopy was used to reveal its composition. The morphological result found that there was a good fiber/matrix adhesion with a 3 wt% fiber loading, as no fiber pullouts or gaps developed between the oil palm fiber and ABS. However, some pores and fiber pullouts were found with a 5 and 7 wt% fiber loading. Next, the rheological results showed that the increment of fiber content (wt%) increased the viscosity. This discovery can definitely be used in the extrusion process for making wire filament for FDM. The shear thinning effect was increased by adding 3, 5, or 7 wt% of oil palm fiber. The non-Newtonian index (n) of the composites increased as the number of shear rates increased, indicating that the fiber loading had a significant impact on the rheological behavior. As the fiber loading increased, the viscosity and shear stress values increased as well. As a result, oil fiber reinforced polymer composites can be used as a feedstock filament for FDM.

## 1. Introduction

Additive manufacturing (AM), also known as three-dimensional (3D) printing, has gradually gained traction in the manufacturing industry [1,2,3]. It can be utilized to print intricately shaped metallic, polymer, and composite items with a great design flexibility. By 2020, the market for AM products and services was estimated to exceed $5 billion [4]. FDM is a wire-filament-based method that is frequently used to create functional parts [5,6,7,8,9,10,11]. The impact of the material qualities and mechanical properties of printed parts is critical for their further expansion into FDM. The filament wire is fed into the print head, which allows for the three-dimensional dispensing of the resulting polymer melts on a platform that is lowered one layer at a time. FDM-printed parts are extremely anisotropic and this has a significant impact on the quality of the printed part [12,13]. This demonstrates that the intermolecular diffusion across the interface of the fused filaments is directly proportional to the strength of a thermoplastic interface within an FDM part. The importance of the bonding quality between consecutive filaments is determined by the printing parameters and the melt viscosity of the filament polymer. Previous researchers, such as McIlroy et al. [14,15], Murphy and Collins [16], and Cicala et al. [17], only looked into the effect of rheology on the FDM printing process. As a result, the rheological behavior of polymer melts during FDM processing is an important feature that needs to be investigated further in order to justify the impact of melt behavior on the printing quality. When manufacturing wire filaments for FDM, the flow behavior of a polymer composite material is critical, especially for custom-made composite materials.

For many years, industrial communities and academic circles have paid increasing attention to people’s views of environmental issues and biodegradable polymer products [18,19,20]. “Rheology” refers to how materials deform and flow under stress. It is a basic concept in chemical engineering and the second-most referenced physicochemical feature in engineering research, after particle size distribution [18]. The term rheology was borrowed from Greek and defined as “flow science”—rheo (ρɛω) means “flowing”. The term was created by Reiner [19], and the definition was recognized by a group of prominent scientists. A capillary rheometer is the most straightforward and oldest instrument that measures a fluid’s flow through a pipe of considerable length relative to its diameter. Earlier in history, the water viscosity was measured by Hagen and Poiseuille [20]. Recently, industrial and academic communities have been paying growing attention to biodegradable polymer materials [21,22,23,24]. The capillary rheometer is currently one of the most commonly used tools for measuring the rheological properties of wood plastic composites with primarily powder fillers. Composites reinforced by fibers with a high aspect ratio, on the other hand, are too large to pass through the capillary die (commonly 1 mm in diameter). Furthermore, when simple premixed composites pass through the capillary cylinder, the natural fibers become entangled and detach from the polymer. Because of these fibers’ low dispersion, the mixes are inhomogeneous, making the rheological measurements incorrect.

Many researchers have combined fibers with thermoplastic into composites, crushed them into pellets, and then used a capillary rheometer to test their rheological properties. Kalaprasad et al. used an extruder to make sisal-reinforced polymer composite granules, glass-reinforced polymer composite granules, and combined glass/sisal-reinforced polymer composites, then used a capillary rheometer to characterize the rheology of those composites [25]. The findings revealed that all three types of composites had pseudoplastic tendencies. The non-Newtonian index (n) of the glass fiber composite was lower than that of a sisal fiber composite with a similar fiber composition (20%). The non-Newtonian index of glass/sisal-reinforced composites reduced as the number of glass fibers increased, whereas the viscosity rose. These results showed that varied glass and sisal fiber morphologies had a considerable impact on rheological properties. Smita et al. [26] used a twin-screw extruder to make sisal/high-density polyethylene composites, and then used a capillary rheometer with the extruder to investigate the fiber composites’ steady-state rheological characteristics. The results revealed that the entirety of each material had pseudoplastic properties, which can be described by the equation of the power law. The non-Newtonian index declined as the fiber loading increased from ten percent to thirty percent. The consistency index of the composites, on the other hand, increased. The capillary rheometer was used to assess the rheological characteristics of low-fiber-content composites (not exceeding 40% of the volume). In addition, during the premix and cutting procedures the fibers were sheared and pulled and the matrix could be degraded, all of which compromised the accuracy of the capillary rheometer’s rheological data. Industries have created natural-fiber-reinforced composite (NFRC) materials. An NFRC is a composite material made up of a polymer matrix and natural fibers. Because of its lightweight qualities, low cost, reduced damage to processing equipment, biodegradability, and relatively good mechanical properties, NFRC is widely employed in a variety of applications [27].

Synthetic fibers are human-made fibers that are generated by chemical synthesis and are further categorized as organic or inorganic based on their composition. Carbon fibers, glass fibers, basalt fibers, and aramid fibers are examples of synthetic fibers used in structural applications. Guijun et al. [28] discovered superior corrosive resistance in rods with a fiber random hybrid mode and a smaller diameter. Kevlar-fiber-reinforced composites have great impact strength and tensile qualities but low compression strength due to their anisotropic nature when compared to glass and carbon fibers [29]. Natural fibers are currently being employed in engineering applications as replacements for synthetic fibers. Today, natural fibers have become famous because of their advantages, such as their low cost, abundant availability, environmental friendliness, low density, and good mechanical properties [30]. Natural fibers such as flax, hemp, jute, coir, kenaf, and wood are utilized for reinforcing thermoplastics. On the other hand, natural fibers’ drawbacks include a short lifetime—albeit with a minimal environmental impact when they degrade—and a limited processing temperature, which limit their performance and application [31]. Recently, there has been increased testing and exploration of the use of NFRC filaments for FDM, as well as increased concern for, and knowledge of, environmentally acceptable materials. However, one of the key difficulties for FDM is the melt flow behavior of NFRC materials. Islam et al. [32] reported with the addition of fiber and Maleic anhydride–grafted polypropylene, the melt flow index showed a downward trend. The quality of printed specimens is affected by the melt flow behavior, which is influenced by the pressure, temperature, and physical qualities, including the melting temperature and rheological behavior [33,34]. Extrusion process parameters such as screw speed and barrel temperature have an effect on the manufactured filaments, according to previous studies. Mohammad et al. [35] conducted rheological tests on kenaf fiber-reinforced recycled plastics and discovered that the presence of kenaf fiber negatively affected the flow characteristics. They also identified an indirect link between the composite’s molecular weight and its viscosity. Their results showed that the higher the fiber content of the composites, the more viscous the melts became. Similarly, Mohammed et al. [27] found that the viscosity of kenaf fiber composites rose as their molecular weight increased and as their loading with kenaf fiber increased. Morphology is defined in biology as the study of the shape and structure of organisms [36]. Recently, many researchers have carried out morphological studies on fiber-reinforced polymer composites. For example, Feng et al. [37] studied the effects of sisal fiber morphology on the rheological properties of fiber-reinforced poly (butylene succinate) composites with the same fiber content. They found that the rheological properties of composites reinforced by fibers with different morphologies differed. Subsequently, Kumar et al. [38] discovered that the surface roughness of HDPE was less than that of LDPE using a scanning electron microscope (SEM).

Extrusion, injection molding, compression, and blow molding are examples of polymer processing techniques that might impact rheological behavior and affect the setting of processing parameters. Conversely, fewer published papers have analyzed the behavior of materials in the extrusion process to produce filaments for FDM, particularly for custom-made composite materials. Thus, the aim of our research is to look at the rheological and morphological properties of oil palm fiber-reinforced acrylonitrile butadiene styrene (ABS) composites used as a feedstock material for fused deposition modeling (FDM).

## 2. Materials and Methods

### 2.1. Materials

The oil palm fiber was obtained from local sources. The fibers were soaked for 2 days in water until the oily surfaces were removed. Then, the dried fiber was immersed in a sodium hydroxide (NaOH) solution for 2 h to extract the undesired soluble cellulose, hemicellulose, pectin, and lignin [39]. In natural-fiber-reinforced polymer composites, poor adhesion between the fiber and matrix is a typical issue. Thus, by removing the cellulose, hemicellulose, pectin, and lignin using an alkali treatment, NaOH was found to improve the tensile and flexural characteristics of the material [40]. Experimental results have shown that the flexural strength of composites made from alkali (NaOH)-treated fibers were much better when compared to those of untreated fiber composites [41]. Sghaier et al. [42] studied the characterization of fibers treated by NaOH and found that morphology structure changes can be enhanced by the interaction between the matrix and fibers in composites. Acrylonitrile butadiene styrene (ABS) is a common thermoplastic polymer material that is typically used for FDM. In this study, the ABS PA-747H was used, and it was supplied by a local vendor that is imported from Chi Mei Corporation, Taiwan. A typical ABS is 15% to 35% acrylonitrile, 5% to 30% butadiene, and 40% to 60% styrene, but these proportions may vary within a relatively wide range [43]. The chemical formula for ABS is shown in Figure 1. Table 1 shows the physical and mechanical properties of ABS PA-747H and oil palm fiber.

### 2.2. Oil Palm Fiber Composite Preparation

Figure 2 shows a flow chart for the preparation of oil-palm-fiber-reinforced polymer composites including alkaline treatment, mixing, hot press molding, and crushing. Figure 3 shows the machine used for producing the oil-palm-fiber-mixed ABS plastic in granule form. The oil palm fiber composite samples were prepared by adding the fiber by 3, 5, and 7 wt% into the ABS matrix. Then, the composite was transferred to the pre-mixing process using a high-speed mixer (Cheso model, Cheso Machinery Pte Ltd., Loyang Way, Singapore) at 3000 rpm for 5 min. The next process was a compression molding process; this technique leads the other molding techniques in terms of cost, material consumption, and production rate [44,45]. For the compression molding process, the hot press machine (Gotech model, Gotech Testing Machines Inc) from Taichung City, Taiwan, was used with parameters set at 180 °C for 20 min under a load of 1 ton. The compound resulting from the hot press process was in plate form. Therefore, it had to be crushed in order to acquire the composite material in granule form. For this purpose, a crushing machine (Cheso model, Cheso Machinery Pte Ltd., Loyang Way, Singapore) was used to produce the composite granules (with 3, 5, and 7 wt% of oil palm fiber composite), which were then ready to be used in the rheology test.

### 2.3. Rheological Test

The rheological measurements were taken using an Instron capillary rheometer model SR20 (Instron, Norwood, MA, USA) at different piston speeds in the range of 0.00024–1200 mm/min, as shown in Figure 4. The capillary used was made of tungsten carbide with a length to diameter (L/D) ratio of 5:1. The samples with 0, 3, 5, and 7 wt% oil palm fiber composite were loaded into the barrel of the extrusion assembly and forced down into the capillary using a plunger. The experiment began by setting the die temperature to 220 °C, followed by 230 °C and 240 °C for each sample. The shear rate values were set to 200, 400, 600, 800, and 1000 s^−1^. After allowing a resting time of 5 min, the melt was extruded through the capillary at predetermined plunger speeds. The initial position of the plunger was kept constant in all of the experiments, and shear viscosities at different shear rates were obtained from a single charge of the material [46]. Referring to the previous study by Son Y, fiber-reinforced polymers are non-Newtonian or known as shear thinning fluids [47]. For Newtonian fluids, the wall shear rate is given by:(1)$$\dot{\gamma }\;=\;\frac{4Q}{\pi r^{3}}$$
where *Q* is the volumetric flow rate and *r* is the capillary radius. However, due to the shear thinning behavior of these composites, a Rabinowitsch correction was made to obtain the true shear rate for the power law model [47]. The following is the power law equation for shear stress:(2)$$\tau =k{\dot{\gamma }}^{n-1}$$
where *τ* is the shear stress, *k* is the consistency index, $\dot{\gamma }$ is the shear rate, and n is the flow behavior index. The parameters *k* and n characterize the rheology of power law fluids. Rabinowitsch correction accounts for the non-Newtonian behavior of the melt in which the apparent shear rate $\dot{\gamma }$_a_ was converted into the true shear rate $\dot{\gamma }$_w_ by the following formula [48]:(3)$${\dot{\gamma }}_{w}=\left[\frac{3n+1}{4n}\right]\;{\dot{\gamma }}_{a}$$

### 2.4. Scanning Electron Microscope (SEM)

The morphology of the compounded oil palm fiber/ABS composite was observed using a scanning electron microscope, model JEOL (JSM 6010PLUS/LV, JEOL Ltd., Tokyo, Japan), using a platinum coating and 20 kV acceleration, as shown in Figure 5.

## 3. Results and Discussion

### 3.1. Rheological Behavior

The purpose of performing rheological tests was to characterize the flow behavior and identify the correct melting temperature (T_m_) of the oil palm fiber composite. Rheology tests were successfully performed for the oil palm fiber/ABS composites with compositions of 0, 3, 5, and 7 wt% at three different die temperatures: 220, 230, and 240 °C. Unfortunately, the composite material was able to flow out only at the 240 °C die temperature, whereas the other temperatures (220 and 230 °C) clogged the die nozzle. As a result, the recommended die temperature for the oil palm composite is 240 °C. This temperature setting was critical for the following step in the fabrication of a wire filament for FDM using the extrusion technique. The extruded samples made using capillary rheometer with a 1.0 mm diameter die are shown in Figure 6. The results show that the color of the 7 wt% sample was much darker compared to that of the 3 and 5 wt% samples. This is due to the higher percentage of oil palm fiber present in the 7 wt% sample. The outer surfaces of the 0 and 3 wt% samples appeared consistent and fine compared to those of the 5 and 7 % wt samples. This means that the surface of extruded samples may become less fine and fragile as a result of the fiber content or loading. Because all of the samples flowed smoothly out of the capillary die, oil palm fiber composite can successfully be fabricated using a twin screw extruder (i.e., for producing wire filament for FDM). Table 2 lists the rheological parameters of the oil palm fiber composite samples including the flow index (n), consistency index (k), and correlation coefficient (R^2^). The values of the flow index for this composite were 0.71, 0.53, and 0.61 for the 0, 3, 5, and 7 wt% samples, respectively. However, the values of the consistency index for the 0, 3, 5, and 7 wt% samples were 0.0004, 0.0012, and 0.0008, respectively. Thus, the trend is that the n value decreased when the fiber loading (wt%) increased. In addition, the values of R^2^ for all samples including the 0, 3, 5, and 7 wt% samples were greater than 0.99. Figure 7 shows the scattering data from a flow curve, including the flow index (n) and correlation coefficient (k), using the power law model. This shows that the flow curve was a satisfactory fit, with a high correlation (R^2^ > 0.99). The flow index of the oil palm fiber composite increased as the number of shear rates increased, indicating that the fiber loading had a significant impact on the rheological behavior.

In rheology, shear thinning is the non-Newtonian behavior of fluids whose viscosity decreases under shear rate. It is sometimes considered synonymous with pseudoplastic behavior [49,50]. Figure 8a shows the graph of the apparent viscosity versus shear rate. Figure 8b shows the corrected viscosity with reference to the Rabinowitsch formula. The viscosity curve for the 5 wt% sample was significantly lower and intercepted the 3 wt% sample’s curve at a 900 s^−1^ shear value. This is due to the inhomogeneous distribution of the fiber in the ABS matrix. The resulting viscosity exhibited shear-thinning behavior when the flow index, n, was less than 1, and the trend of apparent viscosity was decreased with the increment in the shear rate value. Previously, Nair et al. [46] carried out a study on rheological properties of fiber composite, and found that the apparent viscosity of a sisal/PP composite decreased when the shear rate increased. In addition, the non-Newtonian index decreased linearly with an increase in fiber loading from 3 wt% to 7 wt%. Furthermore, the viscosity increased with increasing fiber content (wt%). This is similar to previous research by Mohammad et al. [35], who reported that as the loading of kenaf fiber in composites increased, the viscosity increased, because the molecular weight of the composites increased. Another finding by Qaiser et al. [48] was that the predicted behavior of non-Newtonian shear-thinning increased when the filler content increased the viscosity. The apparent viscosity was estimated to rise with an increasing concentration of fibers at low concentration levels. This was due to the growing number of collisions between particles as they become more closely packed together. Conversely, random packing is no longer achievable at a critical concentration level, and increasing fiber concentration leads to a more ordered anisotropic structure of the fibers in suspension, allowing them to slide easily past one another.

Figure 9 shows the pressure drops measured at a 240 °C die temperature over the shear rate range with three different fiber loadings (vol%). The value of pressure drops declined with a decrease in the fiber content, and the values of viscosity at 5 and 7 wt% flowing through the 1.0 mm diameter dies were the same at the beginning of the stage (with a shear rate of 200–300 s^−1^). As a result, the oil palm composite melts’ apparent viscosities dropped, and they flowed more easily through the capillary die, lowering the pressure drop value. When the content of the PP rose, the pressure drop reduced, according to Meng et al. [51]. The effects of pressure on the shear viscosity of polymer melts have been studied previously [52,53,54,55]. These findings showed that when the pressure of the polymer melt in the die increased, the free volume decreased, which was related to the availability of space between molecules. As a result, the polymer melt’s density increased. As the viscosity of the polymer melts rose, the intermolecular friction increased as well. The viscosity increased as the fiber loading increased. The viscosity value increased from 50.7 (3 wt%) to 56.3 (7 wt%) Pa. s at an 800 s^−1^ apparent shear rate.

Therefore, it is necessary to conduct a rheology characteristics analysis of new materials, such as oil palm fiber composite, in order to solve problems that arise during the extrusion process of producing wire filament for FDM. There are some problems in developing composite feed stock filaments for FDM, such as the process of extrusion damaging the fiber due to exposure to a high temperature and pressure [56]. That will cause nozzle clogging and undesirable results (e.g., over-melting, an inconsistent diameter of wire). Therefore, it is important to determine the characteristics of the fiber composite flow (via rheology analysis) as well as preparation data (e.g., temperature and speed) for the twin screw extruder process. The flow behaviors of polymer and fiber materials are vital for producing filaments for FDM, especially for custom-made composite materials [57].

### 3.2. Morphological Structures

Figure 10 shows SEM micrographs of the oil palm fiber composite samples with fiber loading levels of 0, 3, 5, and 7 wt%. Figure 10b shows that there was a good fiber/matrix adhesion, as no fiber pullouts or gaps appeared between the oil palm fiber and ABS. However, there were some pores and fiber pullouts at 5 and 7 wt% fiber loadings. Figure 10b also indicates that the number of pores and fractured fibers increased with the increase in the fiber content (vol%). This phenomenon was similar to that seen by Hongjie et al. [58], where the increased wood fiber content contributed to a higher number of gaps and fiber pullouts. Figure 9d obviously shows a big pore and clear fractured fiber on the surface. This is due to improper fiber/matrix bonding and the presence of higher fiber loading (7 wt%). The big pores are present because the oil palm fiber started to degrade during the heating process within the temperature range 240–250 °C [59]. Thermal stability can be improved by chemically eliminating a certain proportion of hemicellulose and lignin elements from the fiber. Natural fiber deterioration is a significant challenge in the development of natural fiber composites in composite manufacturing, for example, during the curing process, extrusion process, and injection molding [60,61].

## 4. Conclusions

Rheological tests on the oil palm fiber composite samples were carried out using a capillary rheometer to investigate their rheological behaviors through a capillary die (1.0 mm diameter). The results show that the viscosity of the sample with a fiber loading of 7 wt% was higher than that of samples with loadings of 3 wt% and 5 wt%. It was seen that increasing the fiber content (wt%) increased the viscosity. The apparent viscosity of these composites reduced as the shear rate value increased, indicating pseudoplastic behavior. There was good fiber/matrix adhesion for samples with a 3 wt% fiber loading in terms of morphological behavior, as no fiber pullouts or gaps developed between the oil palm fiber and ABS. However, there were some pores and fiber pullouts in the samples with 5 and 7 wt% fiber loadings. The addition of 3–7% by weight of oil palm fiber increased the shear thinning effect. Furthermore, the pressure drops of the oil palm fiber composite decreased with the decrease in the fiber content. Therefore, data such as viscosity, shear stress, shear rate, and appropriate flow temperature (240 °C) for oil palm composite were collected, and these data will purposefully be used in the next process of making wire filaments and 3D printing processes. Thus, oil-palm-fiber-reinforced polymer composites are suitable for use as a feedstock filament material for FDM.

## Figures and Tables

**Figure 1 polymers-13-03739-f001:**
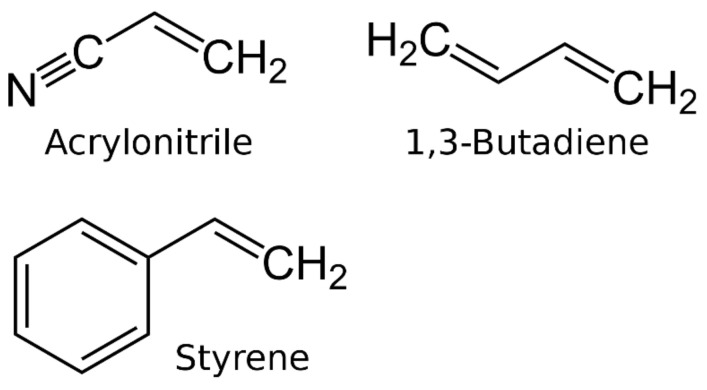
The chemical formula of ABS [43].

**Figure 2 polymers-13-03739-f002:**
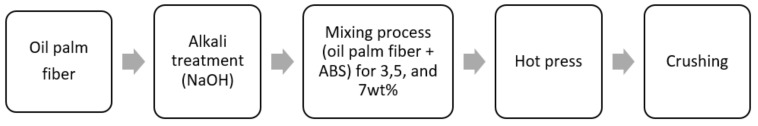
Flow chart for the preparation of oil-palm-fiber-reinforced thermoplastic composites.

**Figure 3 polymers-13-03739-f003:**
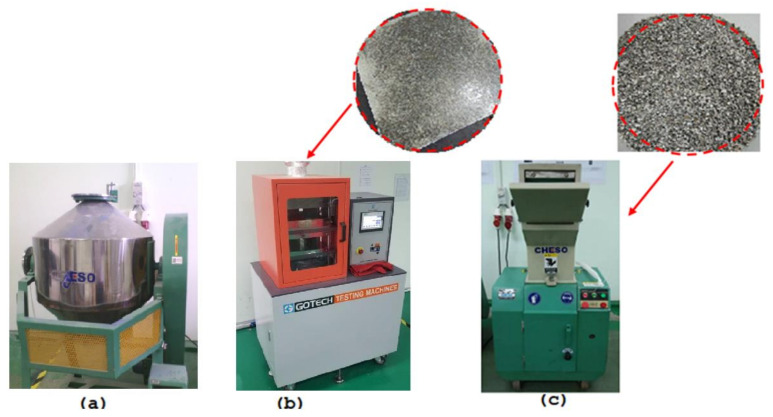
Some of the machines used for producing the oil-palm-fiber-mixed ABS plastic in granule form: (**a**) a mixer, (**b**) a hot pressing machine, and (**c**) a crusher.

**Figure 4 polymers-13-03739-f004:**
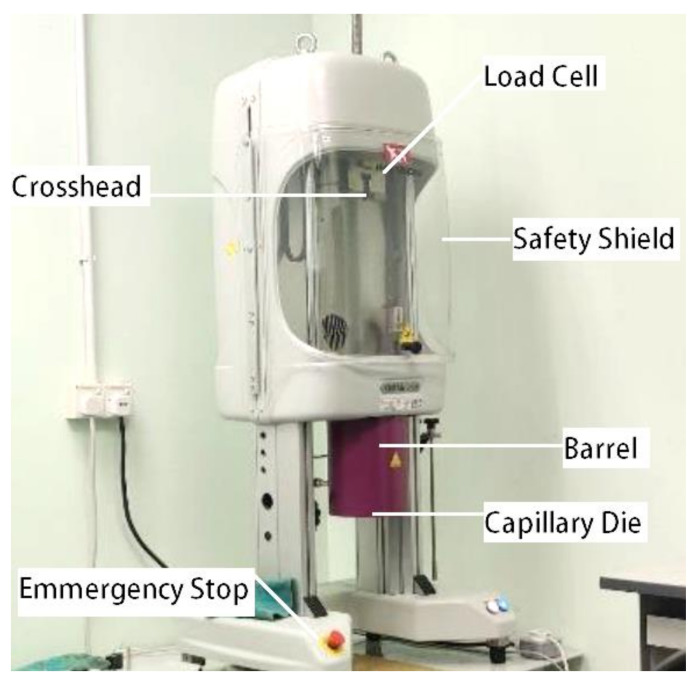
The capillary rheometer used (model Instron SR20).

**Figure 5 polymers-13-03739-f005:**
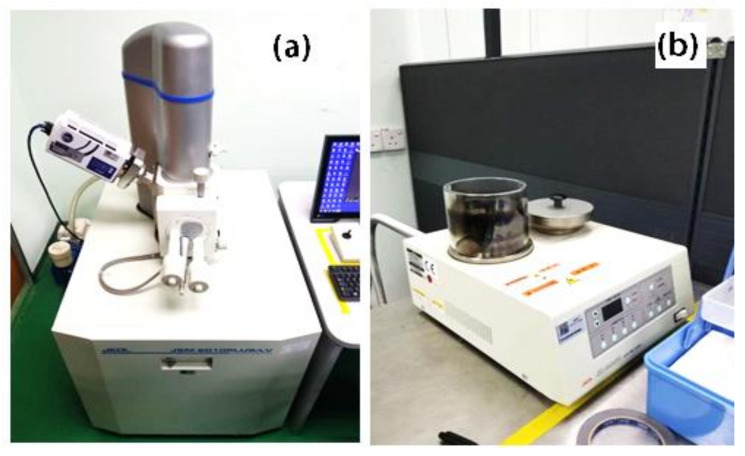
The laboratory setup for the morphology: (**a**) SEM and (**b**) the platinum coating machine.

**Figure 6 polymers-13-03739-f006:**
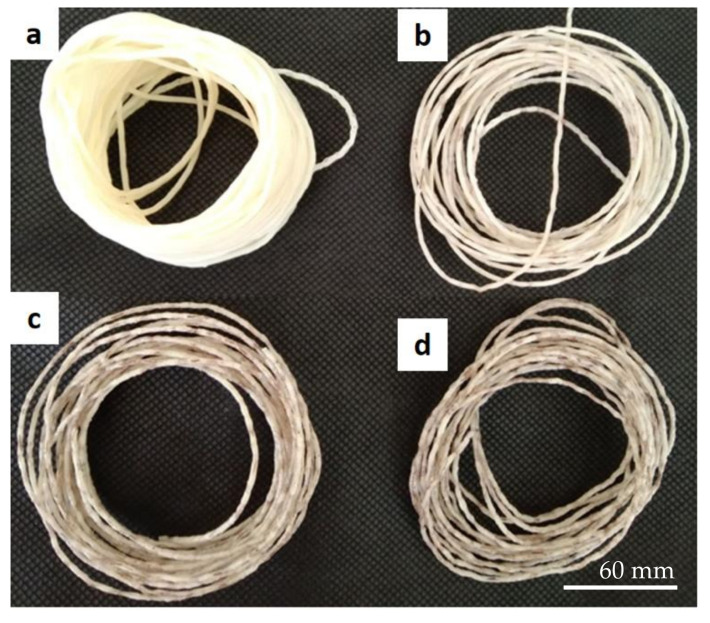
The rheology test samples: (**a**) 0 wt%, (**b**) 3 wt%, (**c**) 5 wt%, and (**d**) 7 wt% of oil palm fiber composite at a 240 °C die temperature.

**Figure 7 polymers-13-03739-f007:**
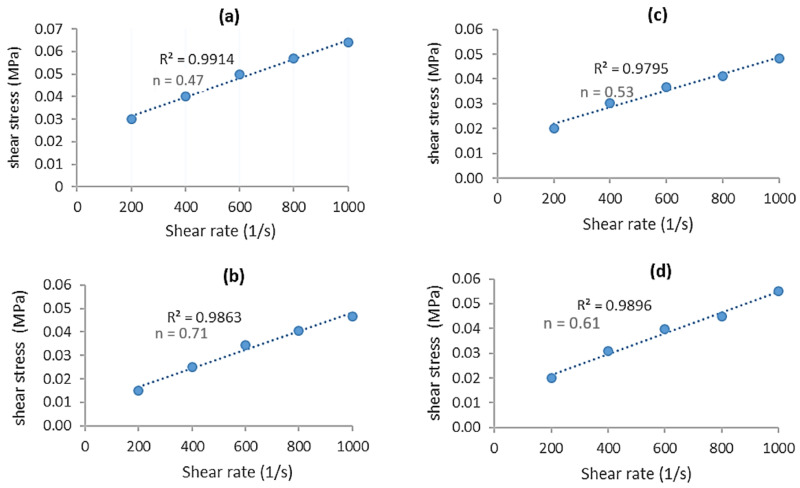
Flow curve graph for the (**a**) 0 wt%, (**b**) 3 wt%, (**c**) 5 wt%, and (**d**) 7 wt% of oil palm fiber composite samples.

**Figure 8 polymers-13-03739-f008:**
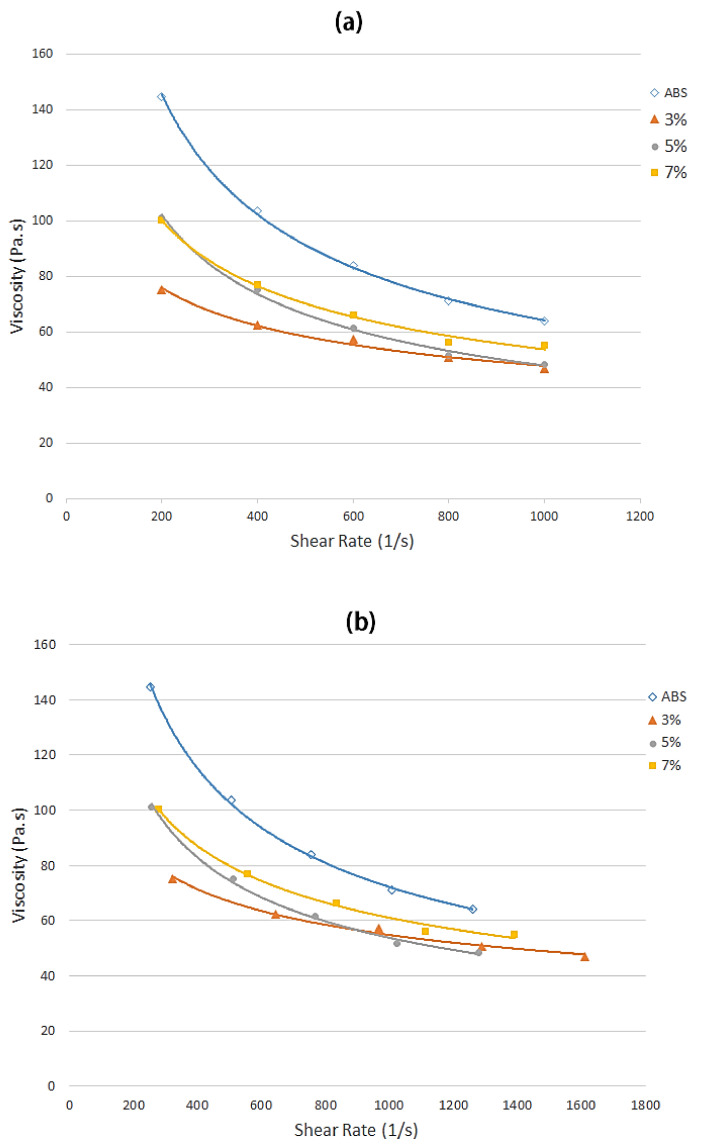
The apparent viscosity of the samples with oil palm fiber composite of 0, 3, 5, and 7 wt% at a 240 °C die temperature: (**a**) the experimental data and (**b**) corrected factor using the Rabinowitsch model.

**Figure 9 polymers-13-03739-f009:**
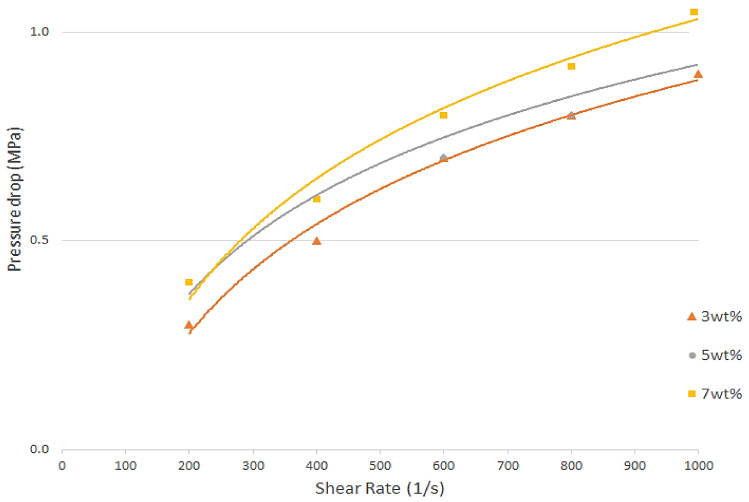
The pressure drop with different fiber loadings at a 240 °C die temperature.

**Figure 10 polymers-13-03739-f010:**
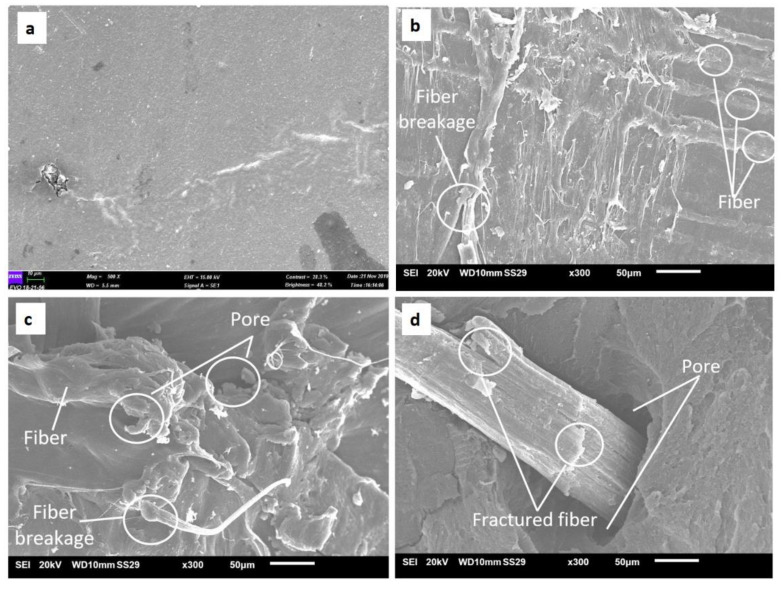
SEM micrographs of the oil-palm-fiber-reinforced ABS composite surfaces: (**a**) 0 wt%, (**b**) 3 wt%, (**c**) 5 wt%, and (**d**) 7wt%. The scale bar in the inset represents 50 μm.

**Table 1 polymers-13-03739-t001:** The physical and mechanical properties of ABS PA-747H and oil palm fiber [39].

Physical and Mechanical Properties	ABS	Oil Palm Fiber
MFI (g/10 min)	13	-
Density (g/cm^3^)	0.9–1.53	0.7
Melting point (°C)	No true melting (amorphous)	-
Cellulose content (%)	-	43–65
Lignin content (%)	-	13–25
Moisture content (%)	-	2.2–9.5
Tensile strength (MPa)	39.0	71.0
Elongation at break (%)	45	11

**Table 2 polymers-13-03739-t002:** Rheological parameter of the power law model.

Fiber Loading (vol%)	Flow Index, n	Consistency Index, k	Correlation Coefficient, R^2^
0	0.4737	0.0024	0.9972
3	0.7114	0.0004	0.9978
5	0.5342	0.0012	0.9953
7	0.6112	0.0008	0.9962

## Data Availability

The data presented in this study are available on request from the corresponding author.

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
