# Peer review of "Rheological and Morphological Properties of Oil Palm Fiber-Reinforced Thermoplastic Composites for Fused Deposition Modeling (FDM)"

_polymers, 2021, doi:10.3390/polym13213739_

Round 1
Reviewer 1 Report
1, The paper "RHEOLOGICAL CHARACTERISTICS OF NATURAL FIBER REINFORCED THERMOPLASTIC COMPOSITES FOR FUSED DEPOSITION MODELING (FDM)" is devoted to the important and actual scientific problem, however the novelty of the reported results remained obscure for me.
The authors concluded that: The results reveal that the in-
creasing pressure of the polymer melt in the die led to a reduction of the free volume; this was related to the availability of space between molecules. As a result, the density of the polymer melt increased. The viscosity of the polymer melts increased with a corresponding growth of the intermolecular friction".
These are well-known scientific facts. The novelty of the reported results should be clearly stated.
2. Scale bars are necessary for Figures 2-5.
3. Statistical scattering of the flow index should be reported in Table 2. I do not think that it is possible establish experimentally the flow index with an accuracy of four significant figures as it is reported in the manuscript.
Author Response
Dear reviewer, Please refer to the attachment for the point-to-point response. Your positive comments are really appreciated. Thank you, Ahmad, M.N.
Reviewer 2 Report
Rheological and morphological properties of natural fiber reinforced thermoplastic composites were studied. The results showed that the increment of fiber content increased the viscosity. As the fiber loading increased, the viscosity and shear stress values increased. Before accepting the paper, the authors should carefully revise the full text according to the following comment. In particular, the introduction was written very poor and lacked of the rigorous logic, which must be further improved. Specific comments are as follows.
- Abstract, ABS appears for the first time and needs to be given its full name. The rheology and morphology of composite were studied. While, the Abstract only gives some rheological results, and does not mention some relevant analysis results of composite morphological Properties. It is suggested to add this part. In addition, the reviewer considers it is abrupt about the impact of fiber loading. How to realize the fiber loading condition? It is recommended to give relevant statements.
- At present, the writing logic of the introduction is very chaotic. Paragraphs 2 to 5 mainly introduce the rheology and related research. It is suggested that the authors combine the above four paragraphs into two paragraphs, mainly including the basic behavior of rheology and relevant research results. In addition, in the paragraph 5, the introduction of natural fiber reinforced polymer composites should be presented separately.
- For the statement, “Natural fibers are currently being employed in engineering applications to replace synthetic fibers. Natural fibers have been preferred over synthetic fibers because of their advantages, such as inexpensive cost, abundant availability, environmental friendliness, low density, and superior strength performance [27].”, the authors mentioned that natural fibers have many advantages over synthetic fibers. However, the statement is not rigorous, especially for superior strength properties (Table 1). As known, synthetic fiber has very excellent mechanical properties, fatigue and corrosive resistances. It is suggested that the authors should first introduce the advantage, properties and applications of synthetic fibers (such as typical carbon fiber, glass fiber, basalt fiber, etc.). Please see the following latest and typical research about the properties and applications of synthetic fibers. Journal of Materials Research and Technology, 2021, 14:2812-2831. Composite Structures, 2021, 255: 112869. Composite Structures, 2020; 246: 112418. Through comparing with synthetic fiber, the advantages of natural fibers should be objectively summarized and evaluated.
- The authors studied the effects of rheology and morphological properties on natural fiber reinforced polymer composites. Furthermore, the rheological properties have been fully summarized. However, the reviewers did not see the summary on the morphological properties of natural fiber reinforced polymer composites. It is suggested that the authors make the corresponding supplements.
- In the 2.1 section, Why do you use NaOH to remove the undesired soluble cellulose, hemicellulose, pectin, and lignin? The above mechanism should be further explained and clarified.
- Figure 1 is not clear. It is recommended to replace it with a high-definition picture.
- In section 2.2, it is suggested that the authors provide a flow chart for the preparation of natural fiber reinforced polymer composites.
- In section 3, the authors should firstly put forward why the rheological test was carried out. Why didn’t the author test and characterize the mechanical and thermodynamic properties? The above two properties are important for composites in the engineering application.
- Figure 5 should be placed in the material and testing sections.
- In Table 2, the content of fiber is 0% ~ 7%. Why does the authors use the word of “fiber loading”? Does this mean that fiber is added to the resin after applying the load? Please provide an explanation.
- In Figure 6, Why does the viscosity of ABS resin decrease after adding the fiber? Please explain this.
Author Response
Dear reviewer, Please see the attachment for the response. Thanks, Ahmad, M.N.
Round 2
Reviewer 1 Report
The paper is publishable.
Reviewer 2 Report
The authors have made sufficient modifications.